# Nitric Oxide Improves Salt Tolerance of *Cyclocarya paliurus* by Regulating Endogenous Glutathione Level and Antioxidant Capacity

**DOI:** 10.3390/plants11091157

**Published:** 2022-04-25

**Authors:** Yang Liu, Yichao Yuan, Zhuoke Jiang, Songheng Jin

**Affiliations:** 1Jiyang College, Zhejiang A&F University, Zhuji 311800, China; yuanyichao0217@gmail.com (Y.Y.); jiangzhuoke@gmail.com (Z.J.); 2Zhejiang Provincial Key Laboratory of Resources Protection and Innovation of Traditional Chinese Medicine, Zhejiang A&F University, Hangzhou 311300, China

**Keywords:** antioxidant capacity, *Cyclocarya paliurus*, nitric oxide, NO donors, salt stress

## Abstract

*Cyclocarya paliurus* is commonly used to treat diabetes in China. However, the natural habitats of *C. paliurus* are typically affected by salt stress. Previous studies showed that nitric oxide (NO) level was related to salt tolerance of *C. paliurus*, and its synthesis was induced by exogenous hydrogen sulfide. However, the effects of different NO donors in alleviating the negative effect of salt stress are still unclear. In the present study, *C. paliurus* seedlings pretreated with three NO donors (S-nitroso-N-acetylpenicillamine, SNAP and S-nitrosoglutathione, GSNO and sodium nitroprusside, SNP) were exposed to salt stress, and then, the total biomass, chlorophyll fluorescence parameters, NO and glutathione levels, oxidative damage, and antioxidant enzyme activities were investigated. The results showed that pretreatment of NO donors maintained chlorophyll fluorescence and attenuated the loss of plant biomass under salt stress, and the best performance was observed in *C. paliurus* under SNP treatment. We also found that pretreatment of NO donors further increased the endogenous NO content and nitrate reductase (NR) activity compared with salt treatment. Moreover, pretreatment with NO donors, especially SNP, alleviated salt-induced oxidative damage, as indicated by lowered lipid peroxidation, through an enhanced antioxidant system including glutathione accumulation and increased antioxidant enzyme activities. The supply of NO donors is an interesting strategy for alleviating the negative effect of salt on *C. paliurus*. Our data provide new evidence contributing to the current understanding of NO-induced salt stress tolerance.

## 1. Introduction

It has been estimated that up to 70% of plant growth can be impacted by environmental stress, including drought, high salinity, heavy metal exposure, high or low temperature, and light levels [1,2,3]. Salinity is a crucial factor affecting plant growth and metabolic responses throughout the world [4]. In addition to direct ion damage, salt stress can produce secondary damage to plants through the production of reactive oxygen species (ROS) and osmotic stress [2]. When the salt content in the soil is significantly higher than the optimal concentration for plant survival, it causes a series of stress responses in plants, including the maintenance of osmotic balance and activation of the antioxidant system [5]. In recent years, a great deal of research has been conducted to explore potential ways to improve saline–alkali land, including the breeding of salt-tolerant varieties, the discovery of salt-tolerant genes, and the application of hormones and growth regulators [6,7].

Current studies have shown that nitric oxide (NO) in plants is mainly produced through two pathways, the NO synthase (NOS) and nitrate reductase (NR) pathways [8,9], and is further involved in a series of physiological and biochemical reactions, including root morphogenesis [10], seed germination [11], postharvest quality [12], and the regulation of abiotic stress [13]. Nitric oxide is also involved in the salt tolerance of plant growth. For example, Da Silva et al. [14] reported that salinity-induced accumulation of endogenous NO is associated with modulation of the antioxidant and redox defense systems in *Nicotiana tabacum*. Ren et al. [15] reported that pretreatment with an NO donor alleviates salt stress in seed germination and seedling growth of *Brassica chinensis* by enhancing biochemical parameters and regulating osmolyte accumulation. However, most of the studies focused on sodium nitroprusside (SNP), with few studies on other NO donors. The study of Ahmad et al. [16] showed that another NO donor, S-nitroso-N-acetylpenicillamine (SNAP), could also effectively alleviate the salt-tolerant growth of *Cicer arietinum* by increasing osmolyte accumulation and upregulating of the CAT, SOD, and APX genes.

*Cyclocarya paliurus* (Batal.) Iljinskaja, a multifunctional tree species, is widely distributed in the sub-tropical regions of China [17]. The leaves of *C. paliurus* are rich in health-promoting phenolics and polysaccharide and have long been used as a traditional food in China [18,19,20]. Recently, most of the studies focused on the exploitation and utilization of bioactive substances in *C. paliurus* and the cultivation measures due to its commercial values [21,22], but the mechanism of salt tolerance has received less attention [23,24]. Our previous studies showed that nitric oxide level was related to salt tolerance of *C. paliurus*, and its synthesis was induced by exogenous hydrogen sulfide [23]. However, the effects of different NO donors in alleviating the negative effects of abiotic stresses are still unclear.

Thus, we hypothesized that an exogenous supply of NO donors would alleviate salt-induced oxidative stress in *C. paliurus* through increased endogenous NO and glutathione contents and antioxidant enzyme activities. To test this hypothesis, exogenous NO donors and NaCl were added to *C. paliurus* in this study. The aims of our study were (1) to analyze the role of different NO donors in the salt stress tolerance of *C. paliurus*, and (2) to evaluate the effect of exogenous NO on endogenous NO and glutathione levels and the antioxidant system in *C. paliurus* under salt stress. Chlorophyll fluorescence parameters, biomass accumulation, levels of endogenous NO and glutathione in leaves, and antioxidant systems were evaluated.

## 2. Results

### 2.1. Effects of Salt and NO Donors on Total Biomass and Chlorophyll Fluorescence

Changes in total biomass and chlorophyll fluorescence parameters after treatments are shown in Table 1. Compared with the control, salt treatment significantly inhibited the growth of *C. paliurus* seedlings, which was evident from lower values of total biomass and Fv/Fm (Table 1). In comparison with salt-treated seedlings, total biomass increased by 17%, 5%, and 23% under SNAP, GSNO, and SNP treatments, respectively (*p* < 0.05) (Table 1). Similar effects of SNAP, GSNO, and SNP on Fv/Fm of *C. paliurus* leaves were observed. However, the values of NPQ under SNAP, GSNO, and SNP treatments were not significantly different compared with those under salt treatment, which indicated that the plants might be still impacted by salt stress.

### 2.2. Effects of Salt and NO Donors on the Level of Lipid Peroxidation

The effect of NO donors on salt-induced oxidative stress was investigated in terms of the O_2_^−^ production rate and H_2_O_2_ content, while the membrane damage to cells was analyzed by determining MDA content and the relative electrolyte leakage rate. As shown in Figure 1 and Figure 2, compared with the control, salt treatment significantly increased O_2_^−^ production rate and H_2_O_2_ content, MDA content, and relative electrolyte leakage rate of *C. paliurus* by 263%, 72%, 84%, and 29%, respectively (*p* < 0.05). However, pretreatment with the NO donors resulted in significant decreases in O_2_^−^ production rate, H_2_O_2_ and MDA contents and relative electrolyte leakage rate compared with the salt-treated plants (*p* < 0.05), and best performance was observed under SNP treatment. As shown in Figure 1 and Figure 2, compared with salt treatment, O_2_^−^ production rate, H_2_O_2_ and MDA contents, and relative electrolyte leakage rate under SNP treatment decreased by 61%, 37% and 37%, and 15%, respectively (*p* < 0.05).

### 2.3. Effects of Salt and NO Donors on the Endogenous NO Content and NR Activity

Figure 3 shows that salt treatment significantly increased the endogenous NO content and NR activity of *C. paliurus* by 35% and 308% in comparison to the control, respectively. However, pretreatment with NO donors (SNAP, GSNO, or SNP) before salt treatment resulted in a further increase in the NO content (26%, 20%, and 63%, respectively) and NR activity (181%, 54%, and 199%, respectively) in *C. paliurus* leaves compared with the salt-treated plants (*p* < 0.05) (Figure 3).

### 2.4. Effects of Salt and NO Donors on the Endogenous GSH Content and GR Activity

Figure 4 illustrates the endogenous GSH content and the activity of the key enzyme involved in GSH biosynthesis under different treatments. As shown, compared with the control, salt treatment increased the endogenous GSH content and GR activity of *C. paliurus* by 24% (*p* < 0.05) and 31% (*p* > 0.05), respectively. However, pretreatment with NO donors (SNAP, GSNO, or SNP) before salt treatment resulted in a further increase in the GSH content (10%, 2%, and 42%, respectively) and GR activity (122%, 51%, and 426%, respectively) in *C. paliurus* leaves relative to the salt-stressed plants (*p* < 0.05) (Figure 4).

### 2.5. Effects of Salt and NO Donors on Antioxidant Enzyme Activities

As shown in Figure 5A–D, the activities of the studied antioxidant enzymes significantly increased under salt treatment compared with the control plants, including SOD (by 29%), CAT (by 52%), and APX (by 793%), respectively (*p* < 0.05). Moreover, pretreatment with NO donors (SNAP, GSNO, or SNP) before salt treatment further increased SOD (by 22%, 9%, and 27%, respectively), CAT (by 83%, 59%, and 270%, respectively), and APX (by 125%, 69%, and 162%, respectively) activities compared with the salt treatment. However, no significant difference was observed in POD activity among these treatments despite the SNP treatment (Figure 5B). Thus, these results suggested that the modulation of antioxidant enzyme activities by exogenous NO donors, especially by SNP, might alleviate the ROS-triggered oxidative damage caused by salt stress.

## 3. Discussion

In recent years, studies have shown that NO is involved in plant morphological development, root growth, photosynthesis, and the regulation of antioxidant enzyme systems especially when plants are subjected to environmental stress [25,26], but few studies have focused on the effects of different NO donors on plants [27,28,29]. Our previous study showed that the NO level was related to the salt tolerance of *C. paliurus* and it was induced by exogenous H_2_S [23]. This study further confirmed that NO is involved in the salt-tolerant growth of *C. paliurus*, and that different NO donors have quite different effects. This study showed that salinity inhibited the growth of *C. paliurus* seedlings, A was indicated by lower biomass accumulation and decreased Fv/Fm but significantly higher NPQ (Table 1). However, pretreatment with the NO donors alleviated this phenomenon and the best performance was observed under SNP treatment. These results suggest that pretreatment with NO donors is a simple and effective measure to reduce the effect of salt on the growth of *C. paliurus*. These findings are consistent with previous reports on *Crocus sativus* [30], *Arabidopisis thaliana* [31], and *Glycine max* [32].

Chlorophyll fluorescence parameters are often used to reflect the photosystem changes of plants in the face of environmental stresses; the measurement is convenient, accurate, and sensitive [33]. Generally, plants grown under abiotic stress have lower Fv/Fm than non-stressed plants [34], which is consistent with the performance of *C. paliurus* (Table 1). The significantly higher values of Fv/Fm under treatments with NO donors compared with those under salt conditions indicated that pretreatment with NO donors can effectively maintain photochemical efficiency in *C. paliurus* seedlings. Another indicator, NPQ, often reflects the level of heat dissipation dependent on xanthophyll cycle in PSⅡ complexes of plants [35]. In this study, the NPQ value remained higher in the groups treated with NO donors (Table 1), suggesting that the thermal dissipation was enhanced and that NO donors did not completely reverse the damage caused by salinity.

When plants are stressed, the photosynthetic electron transport system produces a variety of reactive oxygen species (ROS), including O_2_^−^ and H_2_O_2_ [36]. This is consistent with the significantly increased values of O_2_^−^ and H_2_O_2_ in the leaves of *C. paliurus* under salt stress (Figure 1). Furthermore, the accumulation of ROS could lead to a break in the osmotic balance of plant cell membrane and eventually lead to membrane lipid peroxidation [37]. The increase in MDA content often indicates the occurrence of membrane lipid peroxidation, while the change in membrane permeability further leads to an increase in relative conductivity [38]. The results in this study showed that there was a significant correlation (r = 0.931, *p* < 0.0001) between MDA content and membrane permeability. Our results showed that the addition of NO donors could effectively maintain the balance of intracellular and extracellular permeable substances and the stability of cell membranes (Figure 2), which was relatively consistent with the results in *Brassica oleracea* [39], broccoli [16], and cotton [40].

The synthesis of NO increases when plants are subjected to stress, as reflected in this work and in previous research [25,26]. Meanwhile, the results of this study showed that the activity of NR and NO content increased simultaneously, indicating that NR is an important enzyme in the synthesis of NO [9]. NR-mediated NO production has been reported to be involved in plant physiological responses under stress [41]. Studies on stressed wheat seedlings have indicated that the application of NO donors could significantly increase both NR activity and NO content [42]. The different effects of NO donors on NO production can also be seen under the salt treatment of *C. paliurus*, and the effects of SNP and SNAP are significantly higher than that of GSNO (Figure 3). Floryszak-Wieczorek et al. [43] also studied the NO production rates from different NO donors, and found that GSNO could not significantly improve the production of NO. In general, these results indicate that endogenous NO might participate in the activation of stress resistance response mechanisms and determine the salt tolerance of *C. paliurus*. However, the effects of NO donors need to be further confirmed by the tests with NO scavenger addition.

One of the mechanisms through which NO participates in plant stress resistance is the activation of the antioxidant system, including the non-enzymatic and enzymatic antioxidant systems [26,44]. A large number of studies have shown that the content of non-enzymatic substances such as proline, phenolics, and soluble proteins in salt-treated plants can be significantly increased with the participation of NO donors [16,23,44]. The results of this study showed that the GSH content and GR activity of *C. paliurus* seedlings under salt stress were significantly increased after treatment with different NO donors (Figure 4), while SNP treatment still showed the best performance. This result is consistent with previous studies on wheat [45], cotton [46], and chickpea [16]. Our previous study also showed that NO induced by hydrogen sulfide was involved in the regulation of the salt tolerance of *C. paliurus*, which was related to the massive synthesis of different phenolic antioxidants [23]. In general, the accumulation of non-enzymatic substances is beneficial for the tolerance of *C. paliurus* osmotic damage caused by stress.

The production of antioxidant enzymes, such as SOD, POD, CAT, and APX, in plants often keeps ROS in a stable state, and these antioxidant systems become activated when ROS are greatly increased under stress [5]. In this study, SOD, POD, CAT, and APX were also compared in terms of antioxidant activity, as these can remove specific ROS such as O_2_^−^ and H_2_O_2_. O_2_^−^ is dissimulated to H_2_O_2_ by SOD, which is then eliminated by CAT and APX, to produce H_2_O and O_2_ [47]. We observed that compared to salt treatment, the amount of H_2_O_2_ decreased when SOD activity increased in groups of salt with NO donors (Figure 1 and Figure 5). This might have been because the enzymes and non-enzymatic antioxidants studied included APX, POD, and GSH (Figure 4 and Figure 5), as the ascorbate–glutathione (AsA-GSH) cycle (including APX, GR, and GSH) and POD are effective means to mitigate H_2_O_2_ [48,49]. These results also indicated that the increase in the activity of these antioxidant enzymes mediated by NO donors might be due to post-translational modifications (PTMs), such as persulfidation or S-nitrosation [39,50].

## 4. Materials and Methods

### 4.1. Plant Materials, Growth Conditions, and Treatments

Seeds of *C. paliurus* were collected from Tonggu in Jiangxi province, China (30°73′ N, 116°47′ E) on 28 October 2020. Seeds were germinated using the method of Fang et al. [17] and then sown in nonwoven containers (8.0 cm diameter, 10.0 cm height), which were filled with a 1.2 kg substrate mixture of soil, perlite, peat, and fowl manure (2:4:2:2, *v*/*v*/*v*/*v*, pH 6.44). In late May 2021, healthy seedlings with close to mean ground diameter (3.0 mm) and seedling height (18 cm) were selected and moved to the greenhouse at Jiyang College of Zhejiang Agriculture and Forestry University, Zhuji, China (29°45′ N, 120°15′ E). The greenhouse environment was controlled under a 16 h/8 h (day/night) photoperiod, with an average temperature of 30/18 °C (day/night) and a relative humidity of 70 ± 5%. The nonwoven containers were placed in plastic pots and trays to prevent NaCl leaching. The organic matter content and the total N, P, and K contents in the soil were 73.2 g kg^−1^, 72.4 g kg^−1^, 2.20 g kg^−1^, and 9.5 g kg^−1^, respectively.

Salt treatments were then conducted in late June 2021, and a completely randomized design with three biological replications per treatment and five seedlings per replication was adopted. Seedlings were subjected to one of five treatments consisting of CK (control, distilled water), salt (100 mM NaCl), SNAP (100 mM NaCl with 0.05 mM SNAP (S-nitroso-N-acetylpenicillamine)), GSNO (100 mM NaCl with 0.05 mM GSNO (S-nitrosoglutathione)), and SNP (100 mM NaCl with 0.05 mM SNP). Solutions of NO donors were sprayed over the surfaces of the *C. paliurus* leaves three days before salt treatment (20 mL per seedling). To avoid osmotic shock, NaCl solution was gradually added according to the method of Li et al. [38] in eight steps to achieve a concentration of 0.2%, relative to soil weight. Irrigation was conducted every three days to maintain the field capacity at about 75%. All indices were measured 40 days after treatment when obvious differences were observed.

The expanded mature leaves from similar positions within the mid-portion of the main stem were collected from three replicates of each treatment after analyzing the chlorophyll fluorescence parameters. The leaves of each sample were cleaned and immediately used for the analysis of NO and glutathione levels and enzymatic assays.

### 4.2. Chlorophyll Fluorescence Apparatus

The chlorophyll fluorescence parameters were evaluated after dark adaption for 15 min (based on our previous experiment) [23], using a portable pulse modulation fluorometer (PAM 2500, Heinz Walz GmbH, Effeltrich, Germany). The maximal quantum yield of Pigment System (PS) II (Fv/Fm) was calculated, where Fv was the difference between Fo (the initial fluorescence) and Fm (the maximal fluorescence). Nonphotochemical quenching (NPQ) was determined according to the method of Genty et al. [51].

### 4.3. Determination of Total Biomass

Three intact seedlings from each treatment were harvested 40 days after treatments and the total biomass was measured by oven-drying of the roots and shoots at 80 °C for two days.

### 4.4. Determination of Reactive Oxygen Species (ROS)

The O_2_^−^ production rates of *C. paliurus* leaves under salt and NO treatments were assessed by monitoring the nitrite formed from hydroxylamine in the presence of O_2_^−^, as described by Wang [52]. Frozen leaf material (1 g) was used for each replicate (*n* = 3). Similarly, a frozen leaf (1 g) was used in each replicate (*n* = 3) for the analysis of H_2_O_2_ content in *C. paliurus* leaves, following Patterson et al. [53].

### 4.5. Assessment of Malondialdehyde Content and the Relative Electrolyte Leakage Rate

Lipid peroxidation was measured as the amount of malondialdehyde (MDA) determined by the thiobarbituric acid (TBA) reaction, using 0.3 g of the frozen leaf samples per replicate (*n* = 3) according to Deng et al. [54]. Meanwhile, 10 leaf discs (about 10 mm diameter) from fully expanded leaves of *C. paliurus* were used in each replicate (*n* = 3) to analysis the relative electrolyte leakage rate according to the method of Li et al. [38].

### 4.6. Evaluation of NO and Nitrate Reductase

Fresh leaves (0.3 g) were used in each replicate (*n* = 3) for the analysis of NO content according to Cantrel et al. [55]. NO content was calculated using a standard curve of NaNO_2_ (0–4 μg mL^−1^) and expressed as nmol g^−1^ FW. The activity of nitrate reductase (NR; E.C. 1.6.6.1) was determined according to Jaworski [56]. Similarly, a fresh leaf (0.3 g) was used for each replicate (*n* = 3). The activity of NR was expressed as μmol NO_2_^−1^ g^−1^ FW h^−1^.

### 4.7. Protein Extraction, GSH Content, and Antioxidant Enzymes

Antioxidant enzyme extracts from each treatment were obtained from 0.3 g of frozen leaves according to Yu et al. [57]. The leaves were homogenized at 4 °C in 10 mL of 50 mM phosphate buffer solution (pH 7.8) containing 1% polyethylenepyrrole. The homogenate was then centrifuged at 10,000× *g* at 4 °C for 15 min. The supernatant was collected and used for measurement of GSH content and enzyme activity. Meanwhile, soluble protein concentration was determined according to the method of Bradford [58].

GSH content and glutathione reductase (GR) activity were assayed using kits from the Nanjing Jiancheng Bioengineering Institute (Nanjing, China), following the manufacturers specifications. GR activity was expressed as units g^−1^ protein and one unit was defined as the amount of enzyme depleting 1 mmol NADPH in 1 min. GSH content was expressed as mg GSH g^−1^ protein. SOD activity was analyzed using the method of Beauchamp and Fridovich [59]. SOD activity was expressed as μmol NBT min^−1^ mg^−1^ protein. POD activity was estimated with the method described by Ruan et al. [60]. POD activity was expressed as μmol H_2_O_2_ min^−1^ mg^−1^ protein. CAT activity was measured following the method described by Watanabe et al. [61]. CAT activity was expressed as μmol H_2_O_2_ min^−1^ mg^−1^ protein. Meanwhile, APX activity was measured by the method described by Nakano and Asada [62]. APX activity was expressed as μmol ASC min^−1^ mg^−1^ protein.

### 4.8. Statistical Analyses

One-way analysis of variance (ANOVA) was performed using SPSS 19.0 statistical software (SPSS Inc, Chicago, IL, USA) to evaluate the effects of NO donors and salt treatments following by Duncan’s test (*p* < 0.05). The data are presented as the mean ± standard deviation (SD).

## 5. Conclusions

In the present study, three NO donors (S-nitroso-N-acetylpenicillamine (SNAP), nitrosoglutathione (GSNO), and sodium nitroprusside (SNP)) were added to *C. paliurus* seedlings to analyze the role of NO in the salt stress tolerance of *C. paliurus*. Results showed that exogenous supply of NO donors, especially SNP, increased the endogenous NO accumulation, thereby alleviating salt-induced oxidative damage through enhanced antioxidant activity, including glutathione accumulation and increased antioxidant enzyme activities, thus maintaining chlorophyll fluorescence in plants and attenuating the loss of total biomass. Overall, the supply of NO donors is an interesting strategy for alleviating the negative effect of salt on *C. paliurus*. The present study also provides new evidence contributing to the current understanding of the role of NO signaling in stress tolerance.

## Figures and Tables

**Figure 1 plants-11-01157-f001:**
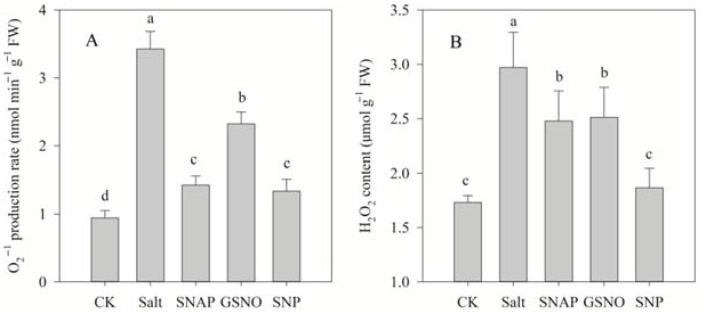
Superoxide radical (O_2_^−^) production rates (**A**) and H_2_O_2_ content (**B**) of *C. paliurus* seedlings under different treatments. CK (control, distilled water), salt (100 mM NaCl), SNAP (100 mM NaCl with 0.05 mM SNAP), GSNO (100 mM NaCl with 0.05 mM GSNO), SNP (100 mM NaCl with 0.05 mM SNP). Bars represent standard deviation (SD) of the means. Different letters suggest significant differences among the treatments at *p* < 0.05 (Duncan’s test).

**Figure 2 plants-11-01157-f002:**
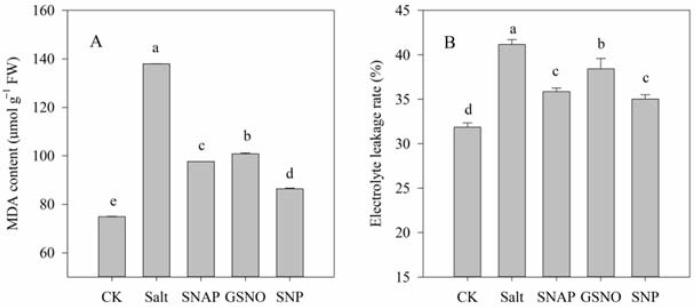
Malondialdehyde (MDA) content (**A**) and the relative electrolyte leakage rate (**B**) of *C. paliurus* leaves under different treatments. CK (control, distilled water), salt (100 mM NaCl), SNAP (100 mM NaCl with 0.05 mM SNAP), GSNO (100 mM NaCl with 0.05 mM GSNO), SNP (100 mM NaCl with 0.05 mM SNP). Bars represent standard deviation (SD) of the means. Different letters suggest significant differences among the treatments at *p* < 0.05 (Duncan’s test).

**Figure 3 plants-11-01157-f003:**
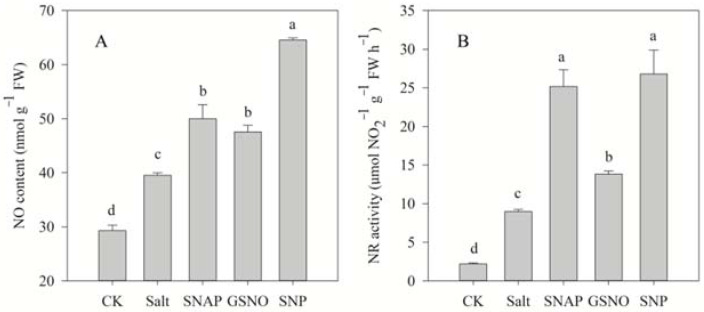
NO content (**A**) and nitrate reductase (NR) activity (**B**) of *C. paliurus* leaves under different treatments. CK (control, distilled water), salt (100 mM NaCl), SNAP (100 mM NaCl with 0.05 mM SNAP), GSNO (100 mM NaCl with 0.05 mM GSNO), SNP (100 mM NaCl with 0.05 mM SNP). Bars represent standard deviation (SD) of the means. Different letters suggest significant differences among the treatments at *p* < 0.05 (Duncan’s test).

**Figure 4 plants-11-01157-f004:**
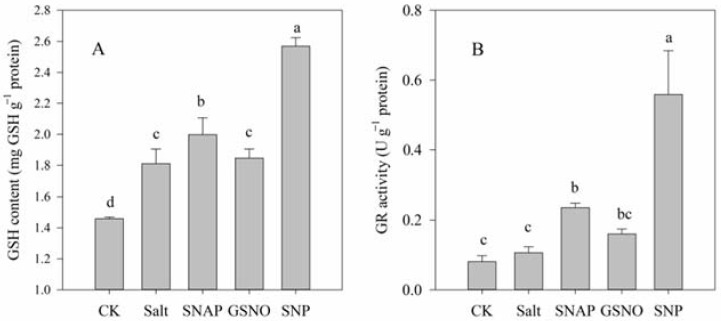
Glutathione (GSH) content (**A**) and glutathione reductase (GR) activity (**B**) of *C. paliurus* leaves under different treatments. CK (control, distilled water), salt (100 mM NaCl), SNAP (100 mM NaCl with 0.05 mM SNAP), GSNO (100 mM NaCl with 0.05 mM GSNO), SNP (100 mM NaCl with 0.05 mM SNP). Bars represent standard deviation (SD) of the means. Different letters suggest significant differences among the treatments at *p* < 0.05 (Duncan’s test).

**Figure 5 plants-11-01157-f005:**
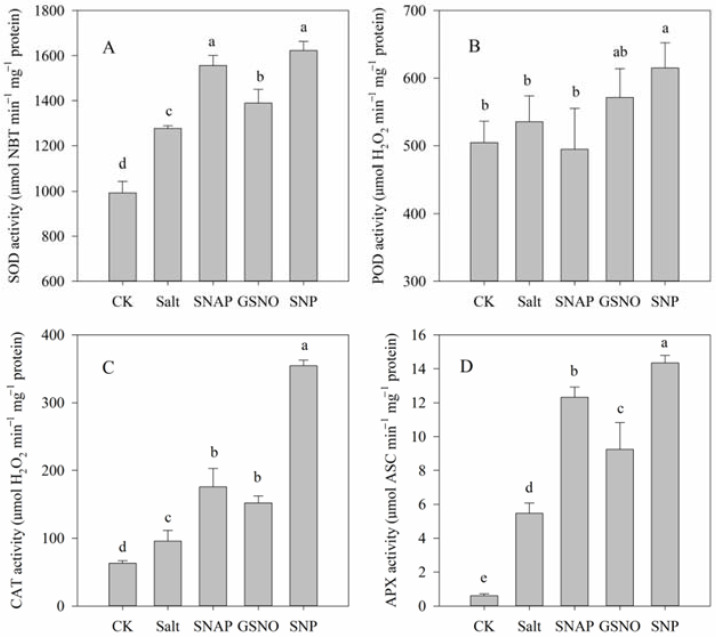
Antioxidant enzyme activities of *C. paliurus* leaves under different treatments. (**A**) SOD; (**B**) POD; (**C**) CAT; (**D**) APX. CK (control, distilled water), salt (100 mM NaCl), SNAP (100 mM NaCl with 0.05 mM SNAP), GSNO (100 mM NaCl with 0.05 mM GSNO), SNP (100 mM NaCl with 0.05 mM SNP). Bars represent standard deviation (SD) of the means. Different letters suggest significant differences among the treatments at *p* < 0.05 (Duncan’s test).

**Table 1 plants-11-01157-t001:** Total biomass and chlorophyll fluorescence parameters of *C. paliurus* in response to different treatments. CK (control, distilled water), salt (100 mM NaCl), SNAP (100 mM NaCl with 0.05 mM SNAP), GSNO (100 mM NaCl with 0.05 mM GSNO), SNP (100 mM NaCl with 0.05 mM SNP). Different letters indicate significant differences (*p* < 0.05) between treatments (*n* = 3, Duncan’s test).

Treatments	Chlorophyll Fluorescence Parameters of *C. paliurus*		
	Total Biomass	Fv/Fm	NPQ
CK	8.61 ± 0.125 ^a^	0.77 ± 0.036 ^a^	0.66 ± 0.032 ^b^
Salt	6.33 ± 0.118 ^e^	0.66 ± 0.016 ^b^	0.78 ± 0.018 ^a^
SNAP	7.40 ± 0.029 ^c^	0.74 ± 0.058 ^a^	0.75 ± 0.046 ^ab^
GSNO	6.66 ± 0.101 ^d^	0.74 ± 0.040 ^a^	0.82 ± 0.073 ^a^
SNP	7.67 ± 0.117 ^b^	0.76 ± 0.008 ^a^	0.75 ± 0.089 ^ab^

## Data Availability

The data presented in this study are available on request from the corresponding author. The data are not publicly available due to privacy.

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
