# Peer review of "Nitric Oxide Improves Salt Tolerance of Cyclocarya paliurus by Regulating Endogenous Glutathione Level and Antioxidant Capacity"

_plants, 2022, doi:10.3390/plants11091157_

Round 1

Reviewer 1 Report

  1. In Table 1, n = 3, but it is not clear what kind of statistical processing was performed for this n number. It is generally considered to be the Mann-Whitney U test, but it is better to clarify it.
  2.  It is compared with SOD, CAT, and APX in terms of antioxidant activity, but these can originally remove only specific ROS. It would be good if there was a description of what active oxygen could be removed.

Author Response

Thank you for your comments. Those comments are all valuable and very helpful for revising and improving our paper, and we have made some revisions based on your suggestion.

(1) In Table 1, n = 3, but it is not clear what kind of statistical processing was performed for this n number. It is generally considered to be the Mann-Whitney U test, but it is better to clarify it.

R:Thank you. One-way analysis of variance (ANOVA) was performed using SPSS 19.0 statistical software (SPSS Inc, Chicago, IL, USA) to evaluate the effects of NO donors and salt treatments following by Duncan‘s test (p < 0.05). And we have added this information in the tables and figures.

(2)  It is compared with SOD, CAT, and APX in terms of antioxidant activity, but these can originally remove only specific ROS. It would be good if there was a description of what active oxygen could be removed.

R:Thank you, we have added this in the discussion section about antioxidant activity. Please check it in the revised MS.

Reviewer 2 Report

Manuscript ID: plants-1691210
Type of manuscript: Article
Title: Nitric oxide improves salt tolerance of Cyclocarya paliurus by 
regulating endogenous glutathione level and antioxidant capacity

This manuscript reports an investigation on the improved salt tolerance of the Cyclocarya paliurus seeds mediated by nitric oxide (three NO donors, S-nitroso-N-acetylpenicillamine, SNAP and nitrosoglutathione, GSNO and sodium nitro-18 prusside, SNP, by ruling the endogenous glutathione level and antioxidant capacity. 

I found the work interesting and well written.
the experiment contains all essential elements to be reproducable
The conclusions are consistent with the work done.

I have only two minor comments, one of style and one a little more substance.
The first concerns the abstract, in particular at the end sentence of the abstract. It would be appropriate to remove the word: in conclusion.

The second consideration concerns the paragraph: Conclusions.
This paragraph, although it contains the summary of the results achieved, it does not introduce the work done, leaving the paragraph unfinished.
I means, that  is missing an opening sentence that summarises the manuscript, that is, the performed work.

I propose to fix these two points, which constitutes a minor revision of the work, before being accepted in the journal PLANTS by MDPI.

Author Response

Thank you for your comments. Those comments are all valuable and very helpful for revising and improving our paper, and we have made some revisions based on your suggestion.

(1) The first concerns the abstract, in particular at the end sentence of the abstract. It would be appropriate to remove the word: in conclusion.

R: Done as suggested.

(2) The second consideration concerns the paragraph: Conclusions.

This paragraph, although it contains the summary of the results achieved, it does not introduce the work done, leaving the paragraph unfinished.

I means, that is missing an opening sentence that summarises the manuscript, that is, the performed work.

R: Thank you. We have added a sentence “in the present study, three NO donors (S-nitroso-N-acetylpenicillamine, SNAP and nitrosoglutathione, GSNO and sodium nitroprusside, SNP) were added to C. paliurus seedlings to analyze the role of NO in the salt stress tolerance of C. paliurus.”to summarise the performed work , please check it in the revised MS.

Reviewer 3 Report

In this paper, the authors demonstrated that nitric oxide improves the salt tolerance of Cyclocarya paliurus, and more precisely, they studied three different nitric oxide donors. Results are clear and well expressed. The introduction was complete.

material and methods:

in general material and methods is more concise.

In 4.1 plant materials, growth conditions, and treatments: please specifier photoperiod and growth temperature

In 4.2 : (based on our previous experiment) please insert the rereference.

In 4.5 Assessment of malondialdehyde content and relative electrolyte leakage rate: please check the reference 54 (Deng et al. 2012).  A better reference is Deng et al. Plant Cell Rep., 30 (2011), pp. 2177-2186.

Minor revisions:

Please add a comma before a conjunction (and /or): lines 140, 141, and 168.

Line 98: please correct “the effect of NO donors was investigated (no were)”.

Line 156: please correct “was observed in POD activity

Author Response

Thank you for your comments. Those comments are all valuable and very helpful for revising and improving our paper, and we have made some revisions based on your suggestion.

(1) In 4.1 plant materials, growth conditions, and treatments: please specifier photoperiod and growth temperature

R: Thank you. We have added this information "the greenhouse environment was controlled under a 16-h/8-h (day/night) photoperiod, with an average temperature of 30/18℃ (day/night) and a relative humidity of 70±5%.” in the revised manuscript.

(2) In 4.2 : (based on our previous experiment) please insert the rereference.

R: Added as suggested.

(3) In 4.5 Assessment of malondialdehyde content and relative electrolyte leakage rate: please check the reference 54 (Deng et al. 2012).  A better reference is Deng et al. Plant Cell Rep., 30 (2011), pp. 2177-2186.

R: Changed as suggested.

(4) Please add a comma before a conjunction (and /or): lines 140, 141, and 168.

R: Done as suggested.

(5) Line 98: please correct "the effect of NO donors was investigated (no were)"

R: Done as suggested.

(6) Line 156: please correct “was observed in POD activity

R: Done as suggested.